# Improving Conditional Coverage via Orthogonal Quantile Regression

**Shai Feldman**
Department of Computer Science
Technion, Israel
shai.feldman@cs.technion.ac.il

**Stephen Bates**
Departments of Statistics and of EECS
UC Berkeley
stephenbates@cs.berkeley.edu

**Yaniv Romano**
Departments of Electrical and Computer Engineering
and of Computer Science
Technion, Israel
yromano@technion.ac.il

## Abstract

We develop a method to generate prediction intervals that have a user-specified coverage level across all regions of feature-space, a property called *conditional coverage*. A typical approach to this task is to estimate the conditional quantiles with quantile regression—it is well-known that this leads to correct coverage in the large-sample limit, although it may not be accurate in finite samples. We find in experiments that traditional quantile regression can have poor conditional coverage. To remedy this, we modify the loss function to promote independence between the size of the intervals and the indicator of a miscoverage event. For the true conditional quantiles, these two quantities are independent (orthogonal), so the modified loss function continues to be valid. Moreover, we empirically show that the modified loss function leads to improved conditional coverage, as evaluated by several metrics. We also introduce two new metrics that check conditional coverage by looking at the strength of the dependence between the interval size and the indicator of miscoverage.

## 1   Introduction

Learning algorithms are increasingly prevalent within consequential real-world systems, where reliability is an essential consideration: confidently deploying learning algorithms requires more than high prediction accuracy in controlled testbeds [1, 2]. Consider, for example, estimating the effects of a drug for a specific person given their demographic information and medical measurements. In such a high-stakes setting, giving a point prediction for the drug's effect is insufficient; the decision-maker must know what the plausible range of effects *for this specific individual*. Instance-wise uncertainty quantification in such settings is critical [3–5]. One approach to this problem comes from the quantile regression and prediction interval literature [6–8]; instead of a point prediction, we can return a range of outcomes that represent the plausible response for a given input. We would like these prediction intervals to achieve a pre-specified coverage level (e.g., 90%) for all inputs—that is, across all regions of feature space. Training models to satisfy this validity guarantee is challenging, however, particularly with complex models like neural networks [9, 10]. In this work, we show how to generate prediction intervals that achieve coverage closer to the desired level evenly across all sub-populations. Technically, we achieve this by augmenting the quantile regression loss function with an additional term that promotes appropriately balanced coverage across the feature space.

35th Conference on Neural Information Processing Systems (NeurIPS 2021).

Formally, consider a regression problem where we are given $n$ training samples $\{(X_i, Y_i)\}_{i=1}^n$, where $X \in \mathbb{R}^p$ is a feature vector , and $Y \in \mathbb{R}$ is a response variable. At test time, we observe a feature vector $X_{n+1}$ and our goal is to predict the unknown value of $Y_{n+1}$ and—importantly—to report on its uncertainty. In this work, we represent this uncertainty by constructing a *prediction interval* $\hat{C}(X_{n+1}) \subseteq \mathbb{R}$ that is likely to contain the response $Y_{n+1}$. In particular, we seek to produce intervals that contain the response with a user-specified probability $1 - \alpha$ that are valid across all regions of feature space:

$$\mathbb{P}[Y_{n+1} \in \hat{C}(X_{n+1}) \mid X_{n+1} = x] \geq 1 - \alpha, \tag{1}$$

a property known as *conditional coverage*. Notice that such prediction intervals are both correct in that they satisfy a coverage guarantee and also are adaptive, in that the size of the prediction intervals can change with the difficulty of the inputs: easy inputs give small intervals and hard inputs give large intervals. Returning to our medical example, consider predicting the outcome of a drug from age, gender, blood pressure, and so on. The conditional coverage requirement in (1) asks that intervals are correct for any age, gender, and health status combination. That is, no matter what an individual's value of the features $x$, the uncertainty quantification must be valid.

The conditional quantiles of $Y \mid X = x$ are the natural way to produce intervals satisfying (1). Let $\alpha_{\text{lo}} = \alpha/2$ and $\alpha_{\text{hi}} = 1 - \alpha/2$. Given the true conditional quantiles, $q_{\alpha_{\text{lo}}}(x), q_{\alpha_{\text{hi}}}(x)$, we can build an *oracle prediction interval* satisfying (1) in the following way:

$$C(x) = [q_{\alpha_{\text{lo}}}(x), q_{\alpha_{\text{hi}}}(x)]. \tag{2}$$

In practice, the conditional quantiles are unknown but can be estimated with quantile regression, yielding the interval $\hat{C}(x) = [\hat{q}_{\alpha_{\text{lo}}}(x), \hat{q}_{\alpha_{\text{hi}}}(x)]$. This approach is attractive because quantile regression yields intervals that are adaptive to heteroscedasticity without requiring parametric assumptions [11–13], but these intervals *might not satisfy the conditional coverage statement* (1), since $\hat{q}_{\alpha_{\text{lo}}}(x), \hat{q}_{\alpha_{\text{hi}}}(x)$ are merely estimations of the true quantiles [14]. Indeed, we observe in experiments 5 that traditional quantile regression often gives intervals with poor conditional coverage, prompting the present investigation.

In this work, we propose a novel regularization scheme to push quantile regression algorithms towards solutions that better satisfies the conditional coverage requirement (1). The core idea is to force the coverage and interval length to be approximately independent, since this independence must hold for the optimal oracle intervals in (2). A method that constructs intervals whose coverage and length are dependent is either sometimes too conservative (generating too wide intervals), sometimes too liberal (yeilding too short intervals), or both. In addition to improved training schemes, we propose two new tools to check the validity of the resulting predictions in a meaningful way. Specifically, in Section 4, we present two new interpretable metrics to asses the violation of conditional coverage, taking advantage of the orthogonality property identified above. We use these (and other) metrics in Section 5 to study our proposal on simulated data and nine real benchmark data sets. We find that our training scheme yields improvements when used together with both a classic [6] and a more recent [15] quantile regression method.

**A synthetic two-group example**

We begin with a small synthetic experiment that demonstrates the challenges of constructing prediction intervals with accurate conditional coverage. We generate a dataset with two unbalanced sub-populations: 80% of the samples belong to a majority group and the remaining 20% to a minority group, where the conditional distribution $Y \mid X$ of the minority group is more dispersed than the majority group. In our experiments, group membership is included as one of the features. See Section S3.1 of the Supplementary Material for a full description of the distribution.

As a baseline method, we first fit a quantile neural network model (`vanilla QR`) by optimizing the pinball loss (see Section 2), attempting to estimate the low $\alpha_{\text{lo}} = 0.05$ and high $\alpha_{\text{hi}} = 0.95$ conditional quantiles. The left panel of Figure 1 shows the coverage obtained by the `vanilla QR` model across training epochs. The coverage on the test data is far lower than suggested by the training data, failing to reach the desired 90% level and increasing as the training progresses. In particular, this gap remains large at the epoch in which the model achieves the minimal loss evaluated on an independent validation set. Here, the empirical test coverage measured over the majority and minority groups is equal to 80% and 68%, and the empirical average lengths evaluated over them are 1.55, 6.45, respectively.

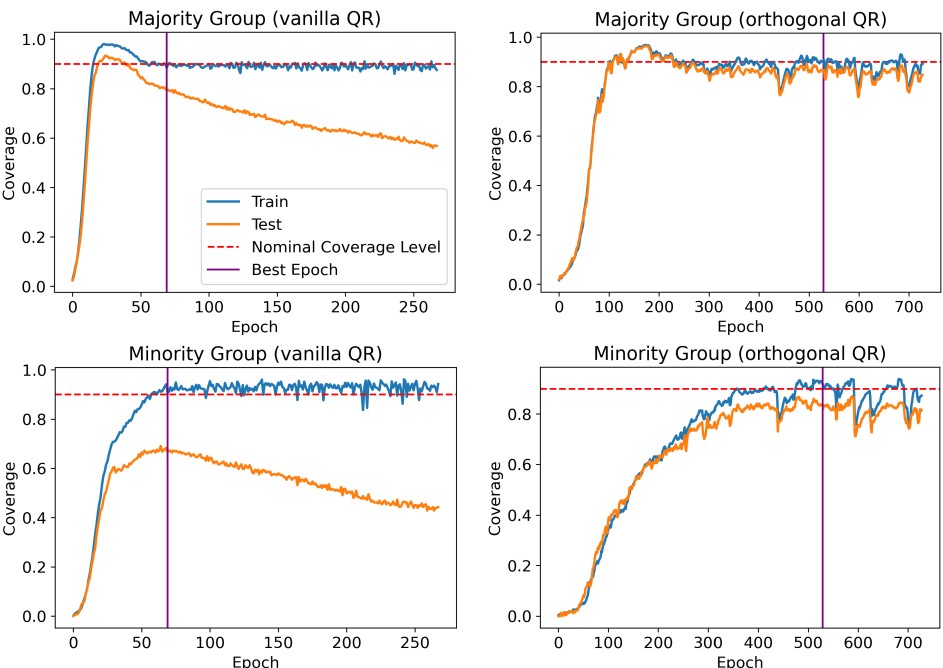

Figure 1: Coverage as a function of the training epochs obtained by quantile natural network model; target coverage rate is 90%. The `vanilla QR` model (left) is trained by optimizing the pinball loss whereas the `orthogonal QR` model (right) is fitted by combining the pinball loss with the proposed independence penalty. The purple vertical line marks the epoch that achieves the smallest empirical risk on a validation set.

Next, we fit our proposed `orthogonal QR` model (which we will formally introduce in Section 3) on the same data. The coverage rate across epochs is illustrated in the right panel of Figure 1. In contrast to `vanilla QR`, the training and testing curves of the majority group overlap for the entire training epochs and both approximately reach the desired 90% level. The minority group coverage has similar behavior, with a significantly smaller gap between the two curves compared to `vanilla QR`. Here, the model that corresponds to the best epoch achieves 86% coverage rate with 1.62 average length on the majority group, and 83% coverage rate with 9.33 average length on the minority group. Importantly, here we are able to train for more epochs before overfitting, which leads to a better final model. To conclude, in this example `orthogonal QR` prevents overfitting and leads to better conditional coverage.

## 2 Preliminaries and related work

### 2.1 Quantile regression

The task of estimating the low and high conditional quantiles, $\hat{q}_{\alpha_{lo}}(x)$, $\hat{q}_{\alpha_{hi}}(x)$, can be expressed as a minimization problem taking the following form:

$$(\hat{q}_{\alpha_{lo}}, \hat{q}_{\alpha_{hi}}) = \operatorname*{argmin}_{f_{\alpha_{lo}}, f_{\alpha_{hi}} \in \mathcal{F}} \quad \frac{1}{n}\sum_{i=1}^{n} \ell_\alpha(Y_i, f_{\alpha_{lo}}(X_i), f_{\alpha_{hi}}(X_i)). \tag{3}$$

Above, $\ell_\alpha$ is a loss function designed to fit quantiles; we discuss two examples next.

The most common loss function for estimating a conditional quantile $\hat{q}_\alpha$ is called the *pinball loss* or check function [6, 9, 16], expressed as

$$\rho_\alpha(y, \hat{y}) = \begin{cases} \alpha(y - \hat{y}) & y - \hat{y} > 0, \\ (1 - \alpha)(\hat{y} - y) & \text{otherwise.} \end{cases} \tag{4}$$

Observe that for the choice $\alpha = 1/2$ the above becomes the $L_1$ norm that is known to estimate the conditional median. By setting $\alpha \neq 1/2$ we get a tilted version of the latter, with a degree controlled

by the value of $\alpha$. In the context of (3), we can set $\ell_\alpha^{\text{pb}}(y, f_{\alpha_{\text{lo}}}(x), f_{\alpha_{\text{hi}}}(x)) = \rho_{\alpha_{\text{lo}}}(y, f_{\alpha_{\text{lo}}}(x)) + \rho_{\alpha_{\text{hi}}}(y, f_{\alpha_{\text{hi}}}(x))$ to simultaneously estimate the low and high quantiles. Throughout this paper, we will refer to this procedure as `vanilla QR`.

A recently-developed alternative to the pinball loss is the *interval score loss* [17], defined as:

$$\ell_\alpha^{\text{int}}(y, f_{\alpha_{\text{lo}}}(x), f_{\alpha_{\text{hi}}}(x)) =$$
$$(f_{\alpha_{\text{hi}}}(x) - f_{\alpha_{\text{lo}}}(x)) + \frac{2}{\alpha}(f_{\alpha_{\text{lo}}}(x) - y)\mathbb{1}[y < f_{\alpha_{\text{lo}}}(x)] + \frac{2}{\alpha}(y - f_{\alpha_{\text{hi}}}(x))\mathbb{1}[y > f_{\alpha_{\text{hi}}}(x)]. \quad (5)$$

Note that the left-most term encourages short intervals and the two remaining components promote intervals with the right coverage. To improve statistical efficiency, it is recommended by [15] to simultaneously estimate all conditional quantiles by minimizing the following empirical risk function: $\mathbb{E}_{\alpha \sim U[0,1]}[\ell_\alpha^{\text{int}}(\cdot)]$.[1] This is the approach we take in our experiments. Note that [15] proposed additional learning schemes for improving efficiency, such as group batching and ensemble learning; see also [18]. These ideas are complementary to our proposal and may further improve the performance of our proposed `orthogonal QR`.

Quantile regression is a large, active area of research, and we finish by pointing out a few representative strands of work in the literature. Estimates of conditional quantile functions using the pinball loss with specific models are proven to be asymptotically consistent under regularity conditions [16]. The work reported in [11–13] offer a non-parametric version of quantile regression. This line of research was further developed by [19] that presented a generalization to additive models that are non-parametric. The pinball loss and interval score loss can be also used to estimate conditional probability distribution [17, 20]. Nevertheless, quantiles can be estimated in other ways rather than minimizing these two loss functions. These include quantile random forest [8] and the method proposed in [21] that iteratively estimates the conditional quantile through the Majorize-Minimize algorithm. Besides the regularization term we propose, there are other suggested penalties that are useful in different situations, such as using sparse modeling in high dimensional response [22–24].

## 2.2 Conformal inference

Conformal inference [25] is a framework for building a prediction intervals that provably attain a weaker *marginal* coverage property:

$$\mathbb{P}[Y_{n+1} \in \hat{C}(X_{n+1})] \geq 1 - \alpha. \quad (6)$$

Importantly, one can guarantee that this holds for any joint distribution $P_{XY}$, sample size $n$, and predictive algorithm. In contrast to (1), the probability statement in (6) is marginal and taken over all the training and test samples $\{(X_i, Y_i)\}_{i=1}^{n+1}$. For example, in the context of the data from Figure 1, intervals that satisfy (6) would be allowed to undercover the minority group and overcover the majority group. Therefore, the statement in (6) is much weaker than that in (1). Yet, the former can be achieved for any distribution whereas the latter can not be achieved for badly-behaved distributions; see [14, 26]. While variants of the guarantee in (6) are possible [27–29], achieving coverage exactly balanced across a continuous feature cannot be done without further assumptions. Much recent work in conformal inference with quantile regression attempts to generate intervals that are adaptive to such heteroscedasticity so that they approximately achieve conditional coverage in (1), while ensuring that $1 - \alpha$ marginal coverage in (6) is exactly achieved [7, 30–35]. The experiments we conduct show that our proposed `Orthogonal QR` method can be used in combination with conformal prediction to improve the conditional coverage property while attaining valid marginal coverage.

## 3 Proposed method: orthogonal quantile regression

### 3.1 Formulating the learning scheme

This section presents a modification of the pinball loss or interval score loss in order to fit models with improved conditional coverage. Denote by $V = \mathbb{1}[Y \in \hat{C}(X)]$ the coverage identifier, and by $L = |\hat{C}(X)|$ the interval's length. Our proposal is motivated by the following observation.

---

[1]The code is hosted at `https://github.com/YoungseogChung/calibrated-quantile-uq`

**Proposition 1** (Independence of width and coverage). *Let $(X, Y)$ be a sample drawn from $P_{XY}$, and let $\mathcal{X}$ be the support of $X$. If the distribution of $Y \mid X = x$ is continuous for all $x \in \mathcal{X}$, and the fixed, deterministic interval-valued function $\hat{C}(X)$ satisfies $\mathbb{P}[Y \in \hat{C}(X) | X = x] = 1 - \alpha$ for all $x \in \mathcal{X}$ and some $\alpha \in (0, 1)$, then the interval satisfies $V \perp\!\!\!\perp L$.*

In particular, the above implies that the length of an interval $L = q_{\alpha_{hi}}(X) - q_{\alpha_{lo}}(X)$ constructed by the true low and high quantiles is independent of the coverage identifier $V$, as stated next.

**Corollary 1.** *Under the assumptions of Proposition 1, an interval constructed by the true conditional quantiles satisfies $V \perp\!\!\!\perp L$.*

We note that an earlier, limited version of this observation appears in [36] in the context conditional coverage for classification problems. All proofs are presented in Section S2 of the Supplementary Material. Since intervals constructed by the true conditional quantiles obey the independence property, forcing the fitted model to approximately satisfy this property during training can result in better conditional coverage for future test points. This leads us to our proposed `orthogonal QR` objective:

$$(\hat{q}_{\alpha_{lo}}, \hat{q}_{\alpha_{hi}}) = \underset{f_{\alpha_{lo}}, f_{\alpha_{hi}} \in \mathcal{F}}{\text{argmin}} \quad \frac{1}{n} \sum_{i=1}^{n} \ell_\alpha(Y_i, f_{\alpha_{lo}}(X_i), f_{\alpha_{hi}}(X_i)) + \gamma \mathcal{R}(\mathbf{L}, \mathbf{V}). \tag{7}$$

where $\ell_\alpha$ is either $\ell_\alpha^{\text{pb}}$ or $\ell_\alpha^{\text{int}}$, $\mathbf{L} \in \mathbb{R}^n$ is a vector that contains $L_i = f_{\alpha_{hi}}(X_i) - f_{\alpha_{lo}}(X_i) = |\hat{C}(X_i)|$ as its elements, and $\mathbf{V} \in \mathbb{R}^n$ is a vector with entries $V_i = \mathbb{1}[Y_i \in \hat{C}(X_i)]$. (To facilitate training with gradient methods, in practice we use a smooth approximation to the indicator function; see Section S1.1 of the Supplementary Material.) The function $\mathcal{R}(\mathbf{L}, \mathbf{V}) \in \mathbb{R}^+$ returns a real-valued score that quantifies the strength of the dependence between $L$ and $V$, where a large value indicates that the two are more dependent; we discuss specific choices in Section 3.2. The regularization strength is controlled by the hyperparameter $\gamma$. In Supplementary Section S4.1 we explain how this parameter is determined, and in Supplementary Section S5.1.3 we demonstrate the effect of this parameter on the performance of our method.

Lastly, we point out that our proposal falls into the broader theme of fitting models while enforcing conditional independence properties, a goal that is important for algorithmic fairness [e.g., 37–39]. This work aims to achieve uncertainty estimates that are equally good across all feature space, a prediction interval analog to the goal of [40].

## 3.2 The orthogonality loss

We now turn to the question of choosing the specific dependence loss penalty, $\mathcal{R}$ in (7). In principle, we could use any dependence measure from the many in the literature: chi-squared tests [41], Pearson's correlation, distance correlation [42], Kolmogorov-Smirnov statistic [43], Randomized Dependence Coefficient [44], Hilbert-Schmidt independence criterion (HSIC) [45], and so on. In this work we focus on Pearson's correlation and HSIC which are described hereafter.

The Pearson's correlation measures the linear dependency between two random variables. Here, the loss is defined as:

$$\mathcal{R}_{\text{corr}}(L, V) = \left| \frac{\text{Cov}(L, V)}{\sqrt{\text{Var}(L)}\sqrt{\text{Var}(V)}} \right|. \tag{8}$$

The advantages of this choice are its simplicity and the minimal computational burden.

Next, HSIC is a more sophisticated, nonlinear complement to the Pearson's correlation measure, which can detect arbitrary complex relationships between the coverage identifier and the interval length. It is an analog of the well-known Maximum Mean Discrepancy (MMD) distance [46], but is a measure of dependence. The idea is that while $\mathcal{R}_{\text{corr}}(L, V) = 0$ does not necessarily imply that $L$ and $V$ are independent, having $\mathcal{R}_{\text{corr}}(g(L), h(V)) = 0$ for every continuous bounded functions $g, h$ guarantees the independence property [47]. While it is impossible to sweep over all possible continuous bounded functions, HSIC offers a tractable solution, guaranteeing that $\text{HSIC}(L, V) = 0$ if and only if $L \perp\!\!\!\perp V$ [45]. In our work, we utilize this measure, and define the orthogonality loss as $\mathcal{R}_{\text{HSIC}}(L, V) = \sqrt{\text{HSIC}(L, V)}$ (taking the square root to magnify small values). This choice is similar to the one advocated in [48].

With these choices of $\mathcal{R}$, we now show that the true conditional quantiles are a solution for the `orthogonal QR` problem.

**Theorem 1** (Validity of orthogonal quantile regression)**.** *Suppose $Y \mid X = x$ follows a continuous distribution for each $x \in \mathcal{X}$, and suppose that $q_{\alpha_{lo}}(X), q_{\alpha_{hi}}(X) \in \mathcal{F}$.*

*Consider the infinite-data version of the* `orthogonal QR` *optimization in* (7)*:*

$$\underset{f_{\alpha_{lo}}, f_{\alpha_{hi}} \in \mathcal{F}}{\operatorname{argmin}} \quad \mathbb{E}\big[\ell_{\alpha}(Y, f_{\alpha_{lo}}(X), f_{\alpha_{hi}}(X)) + \gamma \mathcal{R}\big(|\hat{C}(X)|, \mathbb{I}[Y \in \hat{C}(X)]\big)\big],$$

*where $\hat{C}(X) = [f_{\alpha_{lo}}(X), f_{\alpha_{hi}}(X)]$, $\ell_{\alpha}$ is either $\ell_{\alpha}^{pb}$ or $\ell_{\alpha}^{int}$, and $\mathcal{R}$ is $\mathcal{R}_{corr}$ or $\mathcal{R}_{HSIC}$. Then, true conditional quantiles are solutions to the above optimization problem. Moreover, if the solution is unique for $\gamma = 0$ then the solution is unique for all $\gamma > 0$.*

The uniqueness part of the theorem means that whenever `vanilla QR` is guaranteed to give the correct quantiles in the large-sample limit, then `orthogonal QR` will give the same (correct) solution. The result continues to hold for any dependence measure $\mathcal{R}$ that achieves its minimum value for any two independent variables, a basic property that all dependence measures that we are aware of satisfy. See the proof in Section S2 of the Supplementary Material for further details.

# 4 Metrics for assessing conditional coverage

We next discuss several quantitative measures of conditional coverage. We will introduce two new metrics for conditional coverage, and then review one existing proposal from the literature. Lastly, we will discuss two ad-hoc metrics to help us compare `orthogonal QR` with `vanilla QR` in our upcoming simulation experiments.

## 4.1 Two new metrics for conditional coverage

**Pearson's correlation:** As previewed in the previous section, the Pearson correlation between the interval size and the indicator of coverage (i.e., $\mathcal{R}_{corr}$ from (8)) is a simple, effective way to measure conditional coverage. However, to the best of our knowledge, we are the first to leverage it for this purpose.

**HSIC:** Similarly, we consider the HSIC measure of dependence between the interval size and the indicator of coverage (i.e., $\mathcal{R}_{HSIC}$ above). We estimate this metric as described in [40].[2] As before, to our knowledge this has never been leveraged as a metric to asses conditional coverage.

## 4.2 Other metrics for our empirical evaluations

$\Delta$**WSC:** As an additional measure of conditional coverage, we evaluate the coverage over the worst-slab as proposed in [49].[3] To avoid a case where an improvement in this quantity is obtained by naively enlarging all prediction intervals, we suggest a variant that we call $\Delta$`WSC`. This metric is defined as the absolute difference between the worst-slab coverage and the marginal coverage, both evaluated on test data $\mathcal{I}$:

$$\Delta\text{WSC} = \left| \text{WSC}\left(\{(X_i, Y_i)\}_{i \in \mathcal{I}}; \hat{C}\right) - \text{Coverage}\left(\{(X_i, Y_i)\}_{i \in \mathcal{I}}; \hat{C}\right) \right|.$$

Above, Coverage $\left(\{(X_i, Y_i)\}_{i \in \mathcal{I}}; \hat{C}\right) = \frac{1}{|\mathcal{I}|} \sum_{i \in \mathcal{I}} \mathbb{1}[Y_i \in \hat{C}(X_i)]$ where $\hat{C}(x)$ is a prediction interval method. Importantly, a uniform increase of the length of all intervals will not deceive the $\Delta$`WSC` measure as it will remain fixed.

$\Delta$**ILS-Coverage:** We next consider a measure that checks whether the intervals made larger by `orthogonal QR` compared to `vanilla QR` are necessary for improving the conditional coverage. In general, suppose we are given two algorithms $\mathcal{A}_1$ and $\mathcal{A}_2$ for constructing prediction intervals. Let

$$\Delta L_i = |\hat{C}_{\mathcal{A}_1}(X_i)| - |\hat{C}_{\mathcal{A}_2}(X_i)|$$

---

[2]The code is available at `https://github.com/danielgreenfeld3/XIC`

[3]We used the implementation from `https://github.com/msesia/arc/`

be the difference between the interval length $|\hat{C}_\mathcal{A}(X_i)|$ obtained by $\mathcal{A}_1$ and $\mathcal{A}_2$, evaluated on the same test point $X_i$. Next, let $q_{0.9}(\{\Delta L_i\}_{i \in \mathcal{I}})$ be the 90% empirical quantile of $\{\Delta L_i\}_{i \in \mathcal{I}}$. Then, let ILS be the 10% of samples whose length increased the most:

$$\text{ILS} = \{i : \Delta L_i \geq q_{0.9}(\{\Delta L_i\}_{i \in \mathcal{I}}), i \in \mathcal{I}\}.$$

With this notation in place, we propose the $\Delta\texttt{ILS-Coverage}$ metric:

$$\Delta\texttt{ILS-Coverage} = \left| \text{Coverage}\left(\{(X_i, Y_i)\}_{i \in \text{ILS}}; \hat{C}_{\mathcal{A}_k}\right) - \text{Coverage}\left(\{(X_i, Y_i)\}_{i \in \mathcal{I}}; \hat{C}_{\mathcal{A}_k}\right) \right|.$$

In words, the above is the absolute difference between the coverage over the ILS samples and the marginal coverage, evaluated for each algorithm $\mathcal{A}_k, k = 1, 2$. A smaller value for $k = 1$ indicates that the points with very different size under $\mathcal{A}_1$ and $\mathcal{A}_2$ are handled better by $\mathcal{A}_1$.

$\Delta\textbf{Node-Coverage}$: As a variant of $\Delta\texttt{ILS-Coverage}$, we identify a sub-population character-ized by a small set of features such that the two algorithms $\mathcal{A}_1$ and $\mathcal{A}_2$ produce very different intervals, and check the coverage on this region. To this end, we label the ILS samples as the positive class and fit a binary classifier formulated as a decision tree, aiming to predict whether a sample $X$ belongs to the ILS set. Denote the set of tree nodes in depth at most three that contain at least 5% of the samples by $\{\text{Node}_j\}_j$, where $\text{Node}_j \subseteq \mathcal{I}$. Next, let ND be the set of indices of the samples that belong to the node that maximizes the following ratio: $|\text{Node}_j \cap \text{ILS}| / |\text{Node}_j \setminus \text{ILS}|$. Finally, given a method for constructing prediction intervals $\hat{C}(\cdot)$, compute the distance between the coverage over the ND samples and the marginal coverage, formulated as

$$\Delta\texttt{Node-Coverage} = \left| \text{Coverage}\left(\{(X_i, Y_i)\}_{i \in \text{ND}}; \hat{C}\right) - \text{Coverage}\left(\{(X_i, Y_i)\}_{i \in \mathcal{I}}; \hat{C}\right) \right|.$$

## 5 Experiments

Armed with the performance metrics described in Section 4, we now systematically quantify the effectiveness of the proposed independence penalty when combined with baseline quantile regression methods. In all experiments, we apply a deep neural network as a base model for constructing prediction intervals with $1 - \alpha = 0.9$ coverage level. Section S4 of the Supplementary Material gives the details about the network architecture, training strategy, and details about this experimental setup. Software implementing the proposed method and reproducing our experiments can be found at https://github.com/Shai128/oqr

### 5.1 A synthetic two-group setting

We return to the synthetic two-group setting previewed in Section 1, but first provide more details about the data. In this data set, the difference in distribution between the majority and minority groups is controlled by modifying the noise level of the conditional distribution $Y \mid X$ of the minority group. Furthermore, $X$ has 50 coordinates, the first of which indicates the group membership. Section S3.1 of the Supplementary Material contains more details about the generation of this synthetic data. To analyze the performance of our method, we generate 7000 i.i.d. samples and repeat the following experiment for 30 random train/validation/test splits of the data. We fit a quantile regression model with pinball loss on 5040 training samples, where we tune the number of epochs on an independent validation set that contains 560 samples. The remaining 1400 samples are used to test the model's performance. We pre-processed the feature vector using z-score standardization, and normalized the features and response variables to have a zero mean and a unit variance.

In Table 1 we report the average coverage, length, and conditional coverage metrics for `vanilla QR` and `orthogonal QR` for two minority-group noise levels. (The $\Delta\texttt{ILS-Coverage}$, $\Delta\texttt{Node-Coverage}$ metrics are not reported in this case, since they both essentially correspond to the minority group's coverage level, which is given in the table.) The intervals constructed by the baseline `vanilla QR` undercover both the majority and minority group but; this tendency is more severe for the latter. By contrast, the regularized model achieves similar coverage rates for the two groups, with levels that are closer to the nominal 90%. This is also reflected by the improvement in the `Pearson's correlation`, `HSIC`, and $\Delta\texttt{WSC}$ metrics of conditional coverage. Overall, `orthogonal QR` gives wider intervals for the minority group, which is anticipated since the cov-erage of the baseline model is far below the nominal rate. As for the majority group, our proposed

Table 1: Simulated data experiments. Performance of neural network quantile regression, using either `vanilla QR` (baseline) or `orthogonal QR` (OQR) with penalty term $\mathcal{R}_{\text{corr}}$. The average coverage, length, and percent of improvement in the conditional coverage metrics described in Section 4 are evaluated over 30 independent trials. The standard errors for coverage and length are about 0.25, 0.06, respectively. The standard errors for the conditional coverage metrics are presented in Supplementary Table S9.

| Minority Noise Level | Majority Coverage (%) | Minority Coverage (%) | Majority Lengths | Minority Lengths | Improvement (%) | | |
|---|---|---|---|---|---|---|---|
| | baseline / OQR | baseline / OQR | baseline / OQR | baseline / OQR | `corr` | `HSIC` | $\Delta$`WSC` |
| Low | 79.64 / 87.24 | 66.34 / 82.55 | 1.59 / 1.62 | 6.45 / 9.18 | +64.07 | +53.76 | -13.88 |
| High | 81.07 / 86.97 | 68.74 / 83.74 | 1.88 / 1.71 | 22.05 / 30.11 | +72.21 | +27.99 | +17.96 |

training gives intervals of about the same length compared to the baseline model, while achieving a considerably higher coverage rate. In fact, the regularized model constructs even shorter intervals in the high noise level case. We note that this performance continues to hold even when using interval score loss in place of the pinball loss; see Section S5.1.1 of the Supplementary Material. Lastly, we also examine the effect of our regularization on `weighted QR` [50], and display the results in Section S5.1.2 of the Supplementary Material.

## 5.2 Real data

Next, we compare the performance of the proposed `orthogonal QR` to `vanilla QR` on nine benchmarks data sets as in [15, 30]: Facebook comment volume variants one and two (facebook_1, facebook_2), blog feedback (blog_data), physicochemical properties of protein tertiary structure (bio), forward kinematics of an 8 link robot arm (kin8nm), condition based maintenance of naval propulsion plants (naval), and medical expenditure panel survey number 19-21 (meps_19, meps_20, and meps_21). See Section S3.2 of the Supplementary Material for details about these data sets. We follow the experimental protocol and training strategy described in Section 5.1. We randomly split each data set into disjoint training (54%), validation (6%), and testing sets (40%). We normalized the features and response variables to have a zero mean and a unit variance each, except for facebook_1, facebook_2, blog_data, and bio datasets in which we log transform $Y$ before the standardization.

Table 2 summarizes the performance of `vanilla QR` and `orthogonal QR`. Our proposed method consistently improves the conditional coverage, as measured by the `Pearson's correlation`, `HSIC`, and $\Delta$`WSC` metrics for conditional coverage, even though the latter two are not optimized directly by `orthogonal QR`. In Figure 2, we show the coverage as a function of the interval's length evaluated on the meps_21 data set, and find that `orthogonal QR` (in orange) is closer to the nominal 90% level when compared to the baseline method. Our penalty also improves the $\Delta$`Node-Coverage` in most data sets (see Table 2), indicating that the baseline model tends to undercover the response of at least one sub-

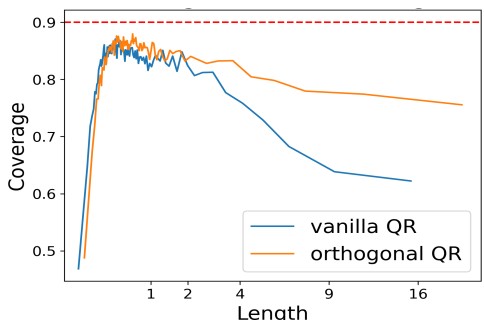

Figure 2: Length versus coverage for `vanilla QR` and `orthogonal QR` over the meps_21 data. In Section S4.3 of the Supplementary Material we explain how the figure was constructed.

population. Turning to the statistical efficiency, observe that the intervals produced by the regularized models tend to be wider than the ones of the baseline method, which is needed to better achieve conditional coverage. We further probe this phenomenon by checking the $\Delta$`ILS-Coverage` which shows that the regions with wider intervals now have better coverage.

In Section S5.2 of the Supplementary Material we provide additional experiments by replacing the pinball loss with the recently-introduced interval score loss. The effect of the decorrelation penalty is similar to the one described above. Moreover, in Section S5.2 we also compare between the decorrelation and HSIC penalties for independence, and show that in most data sets the decorrelation penalty achieves better performance over all metrics of conditional independence at the cost of

Table 2: Conditional coverage on real data. Performance of a neural network model for quantile regression, using either `vanilla QR` (baseline) or `orthogonal QR` (OQR) with penalty term $\mathcal{R}_{\text{corr}}$. The average coverage, length, and percent of improvement in the conditional coverage metrics described in Section 4 are evaluated over 30 independent trials. The standard errors for coverage and width are about 0.45 and 0.03, respectively. The standard errors for the conditional coverage metrics are presented in Supplementary Table S11.

| Dataset Name | Coverage (%) | | Length | | Improvement (%) | | | | |
|---|---|---|---|---|---|---|---|---|---|
| | baseline | OQR | baseline | OQR | corr | HSIC | $\Delta$WSC | $\Delta$ILS | $\Delta$Node |
| **facebook_1** | 88.10 | 90.48 | 1.09 | 1.44 | +81.08 | +76.75 | +33.42 | +65.81 | +47.30 |
| **facebook_2** | 87.38 | 91.13 | 1.07 | 1.41 | +91.27 | +96.36 | +33.86 | +65.71 | +55.63 |
| **blog_data** | 82.89 | 88.92 | 1.36 | 1.64 | +77.49 | +2.27 | +21.44 | +89.94 | +34.41 |
| **bio** | 88.42 | 89.08 | 1.88 | 2.03 | +53.49 | +75.55 | -44.73 | +43.20 | +7.25 |
| **kin8nm** | 84.63 | 88.62 | 0.98 | 1.28 | +27.54 | +48.47 | +14.97 | +57.38 | -16.44 |
| **naval** | 89.89 | 89.50 | 0.56 | 1.49 | +75.19 | +33.98 | -57.52 | +72.36 | -14.34 |
| **meps_19** | 82.44 | 85.16 | 0.84 | 1.00 | +54.16 | +22.25 | +43.28 | +88.05 | +45.43 |
| **meps_20** | 82.81 | 84.27 | 0.86 | 1.03 | +43.51 | +1.99 | +24.06 | +84.71 | +46.88 |
| **meps_21** | 82.50 | 84.07 | 0.86 | 0.99 | +57.46 | +13.62 | +20.78 | +85.02 | +38.28 |

Table 3: Conditional coverage on real data, with conformalization. Performance of a neural network model for conformalized quantile regression, using either `vanilla CQR` (CQR) or `orthogonal CQR` (COQR) with penalty term $\mathcal{R}_{\text{corr}}$. Refer to the caption of Table 2 for further details. The standard errors for coverage and length are about 0.2 and 0.03, respectively. The standard errors for the conditional coverage metrics are presented in Supplementary Table S14.

| Dataset Name | Coverage (%) | | Length | | Improvement (%) | | | | |
|---|---|---|---|---|---|---|---|---|---|
| | CQR | COQR | CQR | COQR | corr | HSIC | $\Delta$WSC | $\Delta$ILS | $\Delta$Node |
| **facebook_1** | 90.08 | 90.11 | 1.09 | 1.43 | +58.28 | +37.61 | +34.86 | +67.79 | +29.46 |
| **facebook_2** | 90.40 | 89.94 | 1.07 | 1.39 | +76.25 | +52.16 | +29.51 | +53.33 | +52.46 |
| **blog_data** | 89.93 | 90.15 | 1.41 | 1.64 | +13.77 | -79.60 | +24.38 | +90.57 | +39.38 |
| **bio** | 90.02 | 89.98 | 1.93 | 2.05 | +51.19 | +71.55 | -30.80 | +39.18 | -40.48 |
| **kin8nm** | 90.11 | 90.11 | 1.12 | 1.33 | +18.16 | +13.06 | -69.52 | +58.14 | -1.44 |
| **naval** | 89.84 | 90.06 | 0.55 | 1.51 | +77.28 | +36.78 | -77.14 | +68.24 | -10.18 |
| **meps_19** | 89.96 | 89.84 | 0.92 | 1.05 | +50.59 | +39.94 | +58.85 | +83.77 | +36.84 |
| **meps_20** | 90.08 | 90.19 | 0.95 | 1.09 | +43.22 | +39.66 | +47.50 | +84.13 | +43.89 |
| **meps_21** | 89.93 | 89.89 | 0.94 | 1.05 | +41.88 | +29.69 | +33.46 | +79.47 | +37.33 |

producing wider intervals. For completeness, we also present the results obtained by quantile regression forests [8] on the real data sets in Supplementary Table S15.

**Conformalized quantile regression results**

In previous experiments the quantile regression methods tend to (marginally) undercover the response variables. This limitation is easily remedied by combining `vanilla QR` or `orthogonal QR` with conformalized quantile regression [30] that adjusts the estimated intervals to exactly achieve the marginal coverage property in (6). Table 3 summarizes the results, demonstrating that by conformalizing the intervals our proposed method precisely achieves the desired marginal coverage while improving the conditional coverage of the baseline model, as measured by the `Pearson's correlation`, `HSIC`, and $\Delta$WSC, $\Delta$Node-Coverage metrics. The two independence metrics indicate that even after after adjusting the intervals to achieve marginal coverage, our method still results in improved independence between the coverage event and length. We note that the $\Delta$WSC, and $\Delta$Node-Coverage metrics have a more muted improvement compared to the setting without conformalization, since the conformalization step smooths out the coverage to some extent. Further details regarding this conformalizing setting is given in Section S4.4 of the Supplementary Material.

## 6  Conclusion

In this work we presented the `orthogonal QR` approach to achieve coverage closer to the desired level evenly across all sup-populations in the setting of quantile regression algorithms. A technical limitation of our method is the use of large batches during training, required to effectively detect the

dependencies between the intervals length and coverage events. In our experiments we focus on i.i.d data, but we believe that the orthogonal loss can be beneficial beyond this, such as in time-series data, which we hope to investigate in future work. A related future direction is to encourage independence between a coverage event and a function of the feature vector, other than that of interval length— similar logic to that of our proposal means that this independence would hold for the true quanties. A clever choice may better capture the relationships between $X$ and the coverage obtained by the model, and further improve conditional coverage.

As a concluding remark, while empirical evidence shows that our `orthogonal QR` approach produces intervals that represent the uncertainty in subgroups more reliably than standard methods, it does not guarantee a valid coverage across all feature space with access to only a finite sample. This guarantee may be necessary for ensuring that predictions are unbiased against a minority group of interest, indexed by an individual's gender or race, for example. To alleviate this, one can combine our methods with the *equalized coverage* framework [28] that builds upon conformal inference to achieve the desired coverage for pre-defined sub-populations, which is a weaker but achievable demand compared to conditional coverage.

## Acknowledgments and Disclosure of Funding

S.F. and Y.R. were supported by the ISRAEL SCIENCE FOUNDATION (grant No. 729/21). Y.R. also thanks the Career Advancement Fellowship, Technion, for providing research support. We thank John Cherian for comments on an earlier version of this manuscript.

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
