# OpenReview forum: "Improving Conditional Coverage via Orthogonal Quantile Regression"
_NeurIPS.cc/2021/Conference — NeurIPS 2021 Poster_

### Official Review · Reviewer_ZNkM · 2021-07-04

**Rating:** 7
**Confidence:** 4

**Summary:**

The paper shows a new loss function that can be used for quantile regression. The key idea is to combine loss functions often used in quantile regression with a new penalty that enforces conditional coverage. Results show improvements according to several metrics.

**Limitations And Societal Impact:**

Yes, limitations have been addressed.


**Main Review:**

The language of the paper is very clear. The ideas and math are also technically correct. Quantile regression is relevant for uncertainty quantification, and thus the contributions are important.

The key to the development of the paper (in particular of the loss function) is Proposition 1, which is neat. Although similar results have been used in other contexts, I have never seen it being used for quantile regression. The empirical results are encouraging.

Major comments:

- How is gamma (the penalty factor) chosen in practice? How sensitive are the empirical results to such a choice? I think this is a very important point that is not addressed in the paper.

- It is natural that the method will outperform others in terms of Pearson and HSIC metrics (because they are explicitly used in the loss function); the improvements with respect to the other metrics are more appealing. However, if gamma is large, wouldn't it be the case that all metrics that were computed in the experiments would be better for OQR (because they basically measure conditional coverage), even though a large gamma does not guarantee that the quantiles would be well estimated? That is, a large gamma enforces good conditional coverage, but by itself, it would not enforce the quantiles to be correctly estimated. Other estimates also satisfy conditional coverage. If this is the case, I feel that other metrics would be necessary to complement the results (such as the pinball loss).

- I also miss comparisons to other quantile estimators that don't directly target the pinball loss, such as quantile regression forests.

Minor comments:

- is the reference to interval score loss "[17]" correct? I think it should be "[15]".

- it would be useful to explain what the worst-slab coverage means

**Time Spent Reviewing:**

10

---

> ### Author Response · Authors · 2021-08-10
> **Reply to reviewer ZNkM**
>
> We appreciate the valuable review, suggestions, and encouragement. In what follows, we address your concerns in detail.
>
> ### Comment 1
>
> > *How is $\gamma$ (the penalty factor) chosen in practice? How sensitive are the empirical results to such a choice? I think this is a very important point that is not addressed in the paper.*
>
> Thank you for raising this point, it was also raised by dPMb, and we repeat the same answer here for convenience. The hyperparameter tuning strategy is described in lines 52-56 in the Supplementary Material; we will include a pointer to that supplementary section immediately after introducing this hyper parameter in Section 3.1. For convenience, we provide this description below.
>
> *The penalty multipliers for each dataset were chosen by an independent train-validation-test split (with seed=42), so the coefficient that achieved the best performance was taken. The multipliers tested for the real data are: 0.1, 0.5, for pinball loss, and 0.1, 0.5, 1, 3, for interval score loss. For the synthetic data we checked: 0.1, 0.5 for both losses. When combining our penalty with pinball loss, the coefficient given to the model is multiplied by 0.1.*
>
> Turning to the sensitivity of the model to the choice of $\gamma$. Thank you for this comment, that was also raised by dPMb. In all experiments the model did not produce trivial intervals as a result of adding the orthogonality loss. To illustrate that, we will include in the manuscript an additional experiment that studies the effect of $\gamma$ (the independence loss multiplier) on the coverage, average length, and variance of the intervals’ length, which we describe in short here. We use the synthetic data from Section 5.1 and choose the pinball loss as the base QR cost function, and found that within the range of values we used for $\gamma$ (0.1 -- 0.5), the constructed intervals are far from being degenerated (i.e., $L=\textrm{constant}$, or $V=\textrm{constant}$ for all samples). In fact, we find that the variance of the length of the intervals is larger (!) than the ones constructed without promoting the orthogonality property.
>
>
> ### Comment 2
>
> > *It is natural that the method will outperform others in terms of Pearson and HSIC metrics (because they are explicitly used in the loss function); the improvements with respect to the other metrics are more appealing. However, if $\gamma$ is large, wouldn't it be the case that all metrics that were computed in the experiments would be better for OQR (because they basically measure conditional coverage), even though a large $\gamma$ does not guarantee that the quantiles would be well estimated? That is, a large $\gamma$ enforces good conditional coverage, but by itself, it would not enforce the quantiles to be correctly estimated. Other estimates also satisfy conditional coverage. If this is the case, I feel that other metrics would be necessary to complement the results (such as the pinball loss).*
>
> This is an interesting point, thank you for bringing this up. This concern is similar to the one raised by dPMb. Note that Proposition 1 states that orthogonality is a necessary criterion for conditional coverage, but the converse is not true. Therefore, if $\gamma$ is large the marginal coverage, $\Delta$WSC and $\Delta$Node-coverage would not necessarily be better for OQR, as these metrics do not rely on the orthogonality property. Indeed, in that case, the Pearson and HSIC metrics would be improved as these reflect the property we promote during training. We will clarify that point in the revised manuscript.
>
> In more detail, the base quantile regression loss function (e.g., pinball loss or interval score loss) is necessary for estimating the conditional quantiles, while the orthogonality loss further guides the model towards a solution of improved conditional coverage property. Minimizing only the orthogonality loss (without the pinball loss) may not lead to accurate conditional quantile estimates, and therefore this may result in intervals of poor conditional coverage. We hope this explains why we don’t entirely agree with the idea that *“a large $\gamma$ enforces good conditional coverage”*. However, the experiments show that encouraging orthogonality in concurrent with minimizing the base quantile regression loss indeed improves conditional coverage. This is indicated by the $\Delta$WSC and $\Delta$Node-coverage metrics. Importantly, as pointed out by the reviewer, we choose these metrics because they are complementary/independent of the loss function used to fit the model, and we view these as objective metrics for conditional coverage. Moreover, the WSC measure is widely used in that literature [1, 2] and fairly interpretable. Since the goal of this work is to improve conditional coverage, we choose to quantify it directly using WSC (among other metrics) rather than comparing the pinball loss or interval score loss on test data, which are hard to interpret.
>
> ### Comment 3
>
> > *I also miss comparisons to other quantile estimators that don't directly target the pinball loss, such as quantile regression forests.*
>
> Thank you for raising this point, we will add the results of quantile regression forests as an additional baseline to the final manuscript.
>
> ### Minor comments
>
> > * *is the reference to interval score loss "[17]" correct? I think it should be "[15]".*
> > * *it would be useful to explain what the worst-slab coverage means*
>
> Thank you for these comments. The interval score loss was proposed in [15] to quantify the quality of the quantile estimates, and was used as a loss function in [17]. We will clarify this as well as include an explanation of the worst-slab coverage to the appendix.
>
>
> References:
>
> [1] Maxime Cauchois, Suyash Gupta, and John C. Duchi. Knowing what you know: valid and validated confidence sets in multiclass and multilabel prediction. Journal of Machine Learning Research, 22(81):1–42, 2021.
>
> [2] Sesia, Matteo and Romano, Yaniv. Conformal histogram regression. arXiv preprint arXiv:2105.08747, 2021.

---

### Official Review · Reviewer_dPMb · 2021-07-16

**Rating:** 5
**Confidence:** 4

**Summary:**

This paper discusses a regularization scheme that aims to improve the conditional coverage of prediction intervals that are learned while performing quantile regression for a pair of quantiles. The empirical experiments show the effect of applying the regularization scheme by demonstrating improvement across a suite of metrics.

**Limitations And Societal Impact:**

I have a couple of concerns and questions regarding the method, and empirical evaluations.

Method:

1. How does the regularization term impact the optimization procedure? I can see multiple ways in which the R(L,V)=0 can be satisfied, e.g. a very wide or narrow prediction interval which always has V=1 or V=0, or any homoscedastic prediction which always satisfies L=constant. Will the regularization term push the converged solution towards these degenerate cases?

2. In relation to the first question, how should $\gamma$ be tuned in (7)?

3. Because of 1), I am dubious of the proposed metrics. E.g. if we only consider a class of homoscedastic models, all models in this class will have R_{corr} and R_{HSIC} = 0?

Empirical evaluation:

4. The evaluation mainly displays how appending the regularization R(L,V) on top of other standard QR losses can improve performance. It's been observed in previous literature that vanilla QR (just optimizing the pinball loss) in the deep learning setting often results in miscalibration, and post-hoc recalibration methods can adjust the quantiles to achieve better calibration. What improvements does orthogonal QR provide over vanilla QR + recalibration? I believe conformalized QR (Section 5.3 and Table 3) is a form of post-hoc recalibration, and I'm not sure if there are significant improvements when comparing the OQR columns in Table 2 with CQR columns in Table 3.


**Main Review:**

This paper is clearly written and well motivated. Learning accurate conditional quantiles and prediction intervals (PIs) is a relevant problem, especially to the general UQ community.

The proposed method is based on the findings presented in Proposition 1, which is applied via a regularization term as stated in equation (7). I believe these are relevant and interesting findings, and provide insights in the problem of learning accurate quantiles and PIs.

I have some questions about the method, which is stated below in the Limitations section.


**Time Spent Reviewing:**

4

---

> ### Author Response · Authors · 2021-08-10
> **Reply to reviewer dPMb**
>
>
> We appreciate your positive feedback and interest in our work. We thank you for the valuable review and suggestions. In what follows, we address your concerns in detail.
>
> ### Comment 1
>
> > *How does the regularization term impact the optimization procedure? I can see multiple ways in which the $R(L,V)=0$ can be satisfied, e.g. a very wide or narrow prediction interval which always has $V=1$ or $V=0$, or any homoscedastic prediction which always satisfies $L=\textrm{constant}$. Will the regularization term push the converged solution towards these degenerate cases?*
>
> Thank you for raising this important point, which is related to the one raised by ZNkM. Indeed, Proposition 1 states that if conditional coverage is satisfied, then the independence property will be satisfied as well. However, the converse is not true, and the example the reviewer provided illustrates that. In other words, $R(L,V)=0$ is not a sufficient condition for achieving conditional coverage. We will clarify this point in the final version of the manuscript. With that said, based on practical evidence we view this concern more of an edge-case rather than the actual behaviour of the method, as discussed below.
>
> Our experience shows that the orthogonality loss tends to increase the variance of the intervals, and we argue that such variability **is desired** when handling heteroscedastic data. In such a case, the conditional distribution of $Y$ given $X$ varies with $X$, therefore the corresponding conditional quantiles must also vary with $X$ to accurately reflect the underlying uncertainty. For example, in the synthetic data, the vanilla QR does not achieve the desired coverage: it constructs too short intervals for the minority group, and too large for the majority. By contrast, our method increases the adaptivity: it widens the intervals of the minority group, and shortens those of the majority group, improving conditional coverage. This shows that an increased adaptivity is achieved by promoting the orthogonality property, which is very much desired in that case. As a result, the variability increases but in a non-trivial way: the increased adaptivity we gain improves conditional coverage.
>
> Our method promotes the orthogonality property in concurrent with minimizing a base quantile regression (QR) loss, e.g., pinball loss or interval score loss.
> Intuitively, the base QR loss shall avoid the trivial cases mentioned above, for which the intervals are trivially wide or narrow (that is, the cases for which $V$ is always 1 or 0) as the quantiles estimates would be of a very poor quality. Importantly, in all experiments (Section 5), we have not encountered the degenerate cases mentioned by the reviewer.
>
> To further illustrate that, we will include in the manuscript an additional experiment that studies the effect of $\gamma$ (the independence loss multiplier) on the coverage, average length, and variance of the intervals’ length, which we describe in short here. In the following experiment, we use the synthetic data from Section 5.1 and choose the pinball loss as the base QR cost function. We find that within the range of values we used for $\gamma$ (0.1 -- 0.5), the constructed intervals are far from being degenerated. In fact, the variance of the length of the intervals is larger than the ones constructed without promoting the orthogonality property (4 for the vanilla QR model and 9.7 for our orthogonal QR method). This, together with the various metrics we use to measure conditional coverage, demonstrate that the orthogonality loss increases the adaptivity of the intervals and improving conditional coverage.
>
> Lastly, we stress that the essence of our proposal is to improve conditional coverage on heteroscedastic data, not homoscedastic. We will clarify this in our manuscript if given the opportunity to revise our manuscript. Using a model that generates homoscedastic predictions on heteroscedastic data, as suggested by the reviewer, would not accurately reflect the underlying uncertainty. Here, the width of the ideal prediction interval should vary with the feature vector $X$, reflecting the variability in the conditional distribution of $Y$ given $X$. Indeed, homoscedastic prediction intervals would be of constant width, independent of the coverage identifier, however, the statistical efficiency and conditional coverage of such a homoscedastic model would be inferior compared to the one obtained by the quantile regression model that we work with. Importantly, we demonstrate that our method improves conditional coverage using an independent performance criteria, such as the one that measures the coverage over the worst-slab. A similar comment was also given by ZNkM.
>
> ### Comment 2
>
> > *In relation to the first question, how should $\gamma$ be tuned in (7)?*
>
> Thank you for raising this point, it was also raised by ZNkM, and we repeat the  remarks here for convenience. The hyperparameter tuning strategy is described in lines 52-56 in the Supplementary Material; we will include a pointer to that supplementary section immediately after introducing this hyper parameter in Section 3.1. For convenience, we provide this description below.
>
> *The penalty multipliers for each dataset were chosen by an independent train-validation-test split (with seed=42), so the coefficient that achieved the best performance was taken. The multipliers tested for the real data are: 0.1, 0.5, for pinball loss, and 0.1, 0.5, 1, 3, for interval score loss. For the synthetic data we checked: 0.1, 0.5 for both losses. When combining our penalty with pinball loss, the coefficient given to the model is multiplied by 0.1.*
>
> ### Comment 3
>
> > *Because of 1), I am dubious of the proposed metrics. E.g. if we only consider a class of homoscedastic models, all models in this class will have $R_\textrm{corr}$ and $R_\textrm{HSIC} = 0$?*
>
> Thank you for bringing up this interesting point. Note that we also evaluate conditional coverage by measuring the coverage over the worst-slab and via the  $\Delta$Node-coverage metric. Both metrics are evaluating conditional coverage in a way that is very different from the orthogonality property, and both are improved by our orthogonal QR approach. This suggests that our improvements are not an illusion, but are robust across different ways of checking conditional coverage. We agree that a homoscedastic model that constructs prediction intervals of a fixed width will have zero $R_{\textrm{corr}}$ and $R_{\textrm{HSIC}}$, however such a model would perform poorly with respect to the worst-slab and via the $\Delta$Node-coverage metrics, and this would be detected in our experiments. This is an important point, and we will add a discussion of this in the final version of the manuscript.
>
> ### Comment 4
>
> > *The evaluation mainly displays how appending the regularization $R(L,V)$ on top of other standard QR losses can improve performance. It's been observed in previous literature that vanilla QR (just optimizing the pinball loss) in the deep learning setting often results in miscalibration, and post-hoc recalibration methods can adjust the quantiles to achieve better calibration. What improvements does orthogonal QR provide over vanilla QR + recalibration?*
>
> Thank you for highlighting this subject. The conformalized QR (CQR) method is indeed a post-hoc calibration that guarantees valid marginal coverage. This stands in contrast with the conditional coverage property that we seek to improve. The CQR correction is not attempting to improve conditional coverage, and even after this calibration step, the conditional coverage rate may be far from  the desired nominal level. For example, suppose we are interested in constructing prediction intervals of 90% coverage, and consider data with two equally-sized groups. CQR adds a fixed constant $Q$ to the interval’s endpoints, and the value of $Q$ is set such that the calibrated interval $C(x) = [q_\textrm{lo}(x) - Q, q_{\textrm{hi}}(x) + Q]$ would achieve the desired marginal coverage rate on test data. If the initial coverage were 70% and 80%, and the target coverage rate were 90%, CQR calibration would increase the initial (uncalibrated) coverage from 70% to 80% for the first group and from 80% to 100% for the second group. As a result, while the marginal coverage will be indeed 90%, however the calibrated procedure still has very bad conditional coverage, since both groups have a coverage level far from the desired 90%.  We see empirically that calibrating orthogonal QR is superior to calibrating vanilla QR in our real-data examples -- see the point below.
>
> > *I believe conformalized QR (Section 5.3 and Table 3) is a form of post-hoc recalibration, and I'm not sure if there are significant improvements when comparing the OQR columns in Table 2 with CQR columns in Table 3.*
>
> Instead of comparing Table 2 to Table 3, for which the base methods used to evaluate the relative performance are different (vanilla QR and CQR), let us focus on Table 3. That table shows that even after calibrating the vanilla QR (called CQR), we get that the calibrated orthogonal QR (called COQR) achieves better conditional coverage, while the marginal coverage is 90%. Lastly, notice that by promoting the orthogonality property we do improve the  $\Delta$WSC and  $\Delta$Node metrics in 6 out of 9 datasets, and Pearson’s correlation, and  $\Delta$ILS metrics for all datasets.

---

### Official Review · Reviewer_CaJn · 2021-07-16

**Rating:** 7
**Confidence:** 4

**Summary:**

The authors propose a novel method to construct prediction intervals with pre-specified conditional coverage probability. Starting point of the paper is the observation that the length $|\hat{C}(X)|$ and conditional coverage indicator $1\{Y \in \hat{C}(X)\}$ are independent whenever the prediction interval $\hat{C}(X)$ has exact coverage probability.  The authors therefore suggest to compute lower and upper bounds of the prediction interval by solving a quantile regression problem augmented by a penalty term that penalizes correlation between $|\hat{C}(X)|$ and $1\{Y \in \hat{C}(X)\}$. The authors compare the performance of their procedure and competing methods on simulated data and nine benchmark data sets.

**Limitations And Societal Impact:**

Yes. They discuss limitations with regard to implementation of their method.

**Main Review:**

I enjoyed reading this paper. It is very clearly written and makes an interesting contribution. Using quantile regression to construct prediction intervals with (asymptotically) correct conditional coverage probability is well-established. Augmenting this traditional approach with a penalty to encourage independence of length and conditional coverage indicator to improve finite sample performance is creative.

My main concern is that the key reason for the poor finite sample performance of the vanilla QR approach might not be due to the violation of the independence property, but simple because of the variability of the quantile regression estimates themselves. Instead of using classical QR, it would be interesting to see whether weighted/ efficient QR (Koenker 2005, Ch. 5.3) already leads to improved conditional coverage. By how much would enforcing the orthogonality condition further improve coverage based on weighted QR?

**Time Spent Reviewing:**

1.5

---

> ### Author Response · Authors · 2021-08-10
> **Reply to reviewer CaJn**
>
> We appreciate your review of our work, and thank you for the helpful suggestions and positive feedback. In what follows we respond to your concerns in detail.
>
> ### Comment 1
>
> > *My main concern is that the key reason for the poor finite sample performance of the vanilla QR approach might not be due to the violation of the independence property, but simple because of the variability of the quantile regression estimates themselves.*
>
> We are not sure what the reviewer means by *“variability of the quantile regression estimates themselves”*, and we would appreciate it if the reviewer would follow-up with further details. Indeed, QR has finite-sample variability, as with all ML methods, but some fitting methods exhibit better finite-sample properties. In this work, we show that between two competing fitting strategies -- quantile regression and orthogonal quantile regression -- the proposed method has better finite-sample conditional coverage properties. This was our main object in this work: improving the conditional coverage of quantile regression.
>
> If the reviewer is pointing to the variance of the constructed intervals, our experience shows that the orthogonality loss tends to increase their variance, and we argue that such variability **is desired** when handling heteroscedastic data. In such a case, the conditional distribution of $Y$ given $X$ varies with $X$, therefore the corresponding conditional quantiles must also vary with $X$ to accurately reflect the underlying uncertainty. For example, in the synthetic data, the vanilla QR does not achieve the desired coverage: it constructs too short intervals for the minority group, and too large for the majority. By contrast, our method increases the adaptivity: it widens the intervals of the minority group, and shortens those of the majority group, improving conditional coverage. This shows that an increased adaptivity is achieved by promoting the orthogonality property, which is very much desired in that case. As a result, the variability increases but in a non-trivial way: the increased adaptivity we gain improves conditional coverage.
>
> ### Comment 2
>
> > *Instead of using classical QR, it would be interesting to see whether weighted/ efficient QR (Koenker 2005, Ch. 5.3) already leads to improved conditional coverage. By how much would enforcing the orthogonality condition further improve coverage based on weighted QR?*
>
> Thank you for raising this point, which we believe improves our work. Following your comment, we implemented weighted QR and fit such a model on the synthetic data from Section 5.1, by setting the minority group noise level to high ($\lambda=10$) in the data generating function. The weights we assign to the samples are the absolute residuals of a regression model fitted to the training set (minimizing the mean squared error). To avoid overfitting, we deploy early stopping, using a validation set. We have examined various options for choosing the weight function, including absolute/squared residuals, and the inverse of the absolute/squared residuals. Another option we tested is to assign weights that balance the majority and minority groups (all minority samples are weighted with a factor of 4, and all majority samples are weighted by 1). Among all options, the absolute residual weighting function achieves the best results, which are described below. In short, our experiment reveals that adding the orthogonality loss to this best version of weighted QR improves conditional coverage as well!
>
> Now, in more detail, the vanilla version of weighted QR results in 91% coverage for the majority group and 71.7% for the minority group. After adding our orthogonality loss, the model achieves coverage of 94.8% for the majority group, and of 85.4% for the minority: the latter is significantly closer to the nominal 90% level compared to the vanilla weighted QR model. Furthermore, for the majority group, we find that the intervals constructed by our orthogonal version of the weighted QR method are smaller than the ones obtained by the vanilla weighted QR. Here, the latter achieves an average interval length of 19.1 for the majority and 24.8 for the minority group. Our approach attains an average length of 17.8 for the majority and 32.3 for the minority group.
> In our revised manuscript, we will include these results and discussion in the final version of the paper as well as apply weighted QR and its orthogonal version to the real data sets.

---

### Decision · Program_Chairs · 2021-09-27

**Decision:**

Accept (Poster)

**Comment:**

This paper proposes a new type of regularization for quantile regression such that the accuracy of finite sample conditional coverage is improved. The proposed regularization is built on an interesting notion about independence between the size of the intervals and the indicator of a miscoverage event. All reviewers agreed that learning accurate conditional quantiles and prediction intervals (PIs) is an important problem in the uncertainty quantification community. The authors are encouraged to consider updating the paper based on the reviewers' comments and suggestions.